# What does social cognition look like in everyday social functioning in Huntington's disease? A protocol for a scoping review to explore and synthesise knowledge about social cognition alongside day-to-day social functioning of people with Huntington's disease

Alexandra Fisher  ,[1,2] Anna Lavis,[1] Sheila Greenfield  ,[1] Hugh Rickards[2,3]

**Correspondence to**
Alexandra Fisher;
alf944@bham.ac.uk

## ABSTRACT

**Introduction** Social cognition is problematic in Huntington's disease (HD). Despite the observations of clinicians and families, there is minimal empirical literature about how it presents in daily life and the impact on social functioning. This protocol forms the basis of a scoping review to synthesise both the quantitative knowledge and qualitative experiences of the HD community so that a visual and narrative map can address what is known and what is not known for the benefit of the community and clinicians and academics alike.

**Methods and analyses** An umbrella scoping review of previous work and a scoping review of newer studies of social cognition and social functioning will be undertaken. The electronic databases PubMed, Medline, PsycINFO, Web of Science, Scopus, Embase and CINAHL will be searched to identify eligible studies from starting from 2003 to June 2023. A grey literature search and grey data search will also be undertaken. Quality appraisal of the included documents will use the Critical Appraisal Skills Programme and Authority, Accuracy, Coverage, Objectivity, Date, Significance checklists. A data charting table will be used for data extraction, with analysis of qualitative data using the framework method. The review findings will be presented in a visual form and in a narrative summary.

**Ethics and dissemination** Ethical review is not usually required as scoping reviews are produced via secondary data analysis, however, this protocol includes the use of grey data from a charity web forum and so in line with best practice for internet mediated research ethical review was sought and approved (STEM Ethical Review Committee, University of Birmingham-ERN_21-1028A). Review findings will be shared with service users and disseminated through a peer-reviewed publications, conference presentations and hosted via the website of the patient association charity the HD Association.

## STRENGTHS AND LIMITATIONS OF THIS STUDY

⇒ This mixed scoping review will amalgamate the Preferred Reporting Items for Systematic Reviews and Meta analyses extended guidelines for the conduct of scoping reviews as well as the Joanna Briggs Institute method for mixed reviews to ensure transparency and rigour.

⇒ Service user and community involvement in the design and conduct of the study which informs this review will ensure that the findings are supported by their experiences, further helping to identify evidence gaps and inform clinical and research practice.

⇒ Grey literature and grey data will be searched to ensure that the review reflects the challenges faced by the Huntington's disease community not currently captured in the scientific literature. A search strategy has been designed to try and reduce the issues associated with the inherent replicability problems by their inclusion.

⇒ While every effort has been made to create a protocol which is inclusive and comprehensive, there are barriers to this defined by the student authors resources that is, the exclusion of non-English Language papers.

## INTRODUCTION

Most of our everyday behaviour is motivated by social and emotional goals.[1] Social cognition with its underlying structures and processes is how we make sense of that social information.

### Social cognition in Huntington's disease

Huntington's disease (HD) is an inherited, neurodegenerative, disease which impacts

communication, cognition, movement, behaviour and neuropsychiatric function.

HD is presently defined as a movement disorder and its clinical diagnosis in gene positive individuals is made following the confirmation of motor onset. However, problems with social cognition and its component skills are now recognised as manifesting earlier in the disease than motor onset.[2 3]

Social cognitive reviews concerning HD have concluded that the processes thought to underlie social cognition such as emotional recognition (ER) expand from problems in recognition of specific emotional valences in restricted sensory modalities to multiple valences and modalities as the disease progresses.[4] Cognitive and affective theory of mind are also impacted.

Nonetheless the extent of this latter knowledge comes from studies predominantly based on brain scanning and tests in controlled rather than everyday settings. Highlighting the setting is important because it is in everyday social interaction that the interrelated processes of social cognition and behaviour combine to produce the social skills which are then translated into social functioning which Kennedy and Adolph[5] define as the embedding of the behaviour over time and in context. Social functioning may include 'occupation, handling of finances and domestic responsibility which encompasses abilities of performing and understanding social negotiations'.[6]

Using this definition as the lens through which we view social functioning in HD, we can see how it contributes to the quality of various activities of daily living in which social interaction is key. Mason *et al*[7] highlight in particular shared decision making in HD as crucial to understand given the nature of the illness and the implications for family and friends. Problematic social functioning in HD seems to be present within crises in intimate[8] and family relationships,[9] early difficulties in work life[10 11] and may ultimately lead to institutionalisation.[12] Furthermore, these challenges have also been observed by family members and clinicians alike.[7]

Despite what we know about social cognition and social functioning, the probable relationship between them has not been brought together to form a cohesive picture of what socially related activity looks like in HD. Addressing this disparity through a synthesis of existing knowledge would allow for clinicians to draw on this understanding by supporting people with HD and their families through education and tackling these issues collectively with policy makers, for researchers to develop more HD sensitive measures vital to be able to deal with gaps in our knowledge, to develop and target earlier therapeutic interventions—all areas which the HD scientific community [13 14] and neuroscience community note they need.

This protocol (and the review which will result from it) form part of the corresponding author's ethnographic study of social cognition in HD. Ethnography is a qualitative method set in the everyday community of people's lives. As part of the consultation with the patient participation involvement (PPI) group of the HD Association

(HDA) for England and Wales for the ethnographic study, the group reflected the value in knowing more about what social cognition looks like in the everyday. Due to its size, the review will be published in several parts in narrative and tabular form. However, the main outcome for the review will be its accessibility to all parties interested in the results and a visual synthesis with key points will available via the HDA's website.

## Why a scoping review?

The motivations for a scoping review are varied[15] but are usually intended to provide an overview and synthesis of a wide body of literature to clarify concepts and identify gaps steering the audience towards more specific questions which then might be addressed in a more focused review or which may be used towards a change in practice or policy.[16]

The benefit of scoping reviews in HD seems to have recently taken hold through the synthesis of literature to aid the definition of irritability[17] but also the 'lived experience' of the illness[18] following genetic testing and these were driven by the mismatch in the literature and what is seen in clinical practice.

Similarly and with direct relevance to this protocol, Gibson and Springer's[19] motivations came from clinical curiosity and they produced a scoping review looking at reasons for behaviour behind the person's 'decreased participation in social activities in comparison to previous social engagement' (which they termed social withdrawal), what the outcome of this was and what interventions had been tried. Their results noted that the heterogeneity of studies and differences in concepts under the 'umbrella' of social functioning led to the meanings and outcomes of studies becoming blurred and so they made a number of recommendations which will be factored into this review. They were also keen to stress that while there are commonalities in social behaviour in HD, future authors should not dismiss the complexities at an individual level which may be fuelling the withdrawal.

While there are already a number of systematic reviews[7 20] and a meta-analysis[4] on social cognition and its proposed underlying components in HD,[3] these now need updating as new knowledge is available. Furthermore, the authors of these reviews recognised that they had not incorporated the associated qualitative and mixed methods on the multiple domains such as work and home life related to social behaviour, interaction and functioning which represent the extension of social cognition in HD; and likewise, the grey literature and grey data which may also refer to these experiences. It is the intention of this multiple method scoping review to incorporate the potential for rich detail from these latter sources.

## What do we mean by grey literature and grey data?

Grey literature is defined as non-commercially published and unpublished literature from organisations in which publishing is not their primary activity and in the case of

health sciences this means that it has not been subject to usual peer review.[21]

Grey data refers to user generated information which is web based such as online forums.

## What is the added value in incorporating grey literature and data in this area?

As Pope[22] stated, the value of broadening the scope of synthesis and including diverse evidence does not mean abandoning rigour but rather judging the evidence in context and answering specific questions rather than intrinsically defining all evidence as lower quality than others.

Paez[23] also suggested the benefits of including grey literature in a systematic review reduced the potential for publication bias, identified current trends and like Pope *et al* considered context which Adams *et al*[24] recommend should now extend to not only 'written' literature such as journals but into the sphere of digital content.

This latter point by Adams is particularly pertinent in the HD area as while the study of social cognition has only relatively recently taken traction, the exploration of everyday social behaviour and functioning in HD has been muted with incongruity between might what appear in the (evidence-based) literature, clinical practice and the discussions in online communities. As Coulson *et al*[25] who looked at the support offered in a HD online community, noted, 'participation in an online support group allows a greater degree of anonymity than face-to-face groups. Such anonymity may facilitate self-disclosure and help individuals in discussing sensitive issues more easily or to give opinions with less fear of embarrassment or judgement than in more traditional face-to-face groups'. It is the contention of the authors of this protocol, in line with Coulson's arguments and our own clinical experiences, that given social functioning in HD maybe fraught with tensions and stigma, investigating this grey data will allow access to stories which illuminate this further.

## The challenges of using grey literature and data

While there is much about grey literature and grey data to be positive about, caution and strategies are needed to limit the potential for what is termed as scope creep but also transparency to enable a credible replicability vital for the perceived rigour of the review.

Scope creep is a term often used in industry but also recognised in research[26] when a project is ill defined and therefore in practice, those who advocate for the inclusion of grey literature and grey data recommend a structured and constrained search with boundaries formed by the aims of the review. A tight set of criteria is also needed to shore up the credible replicability referred to earlier. Replicability is a term used by Hoffecker[21] who proffers it as an alternative to that of the usual reproducibility aimed for in a standard systematic review, arguing that due to the nature of grey information (particularly grey data) reproducibility is a concept that is almost impossible to capture due to the evolving nature of its source that is, an internet forum versus a static record within a database.

## The aims of this review are

1. To explore and summarise the literature related to social cognition in HD.
2. To explore and summarise the literature related to social behaviour and functioning in HD.
3. To draw together the findings using both a mixture of both visual and narrative synthesis in line with current best methodological practice.[27 28]

## PLAN
## Method

To fulfil the aims of this review, the following methodological approaches will be used:

(1) To explore and summarise the literature related to social cognition an umbrella scoping review will be undertaken. As there is already a large amount of systematic work which has reviewed quantitative studies of social cognition in HD, for pragmatic reasons an umbrella style scoping review of these will be presented. An umbrella review is the systematic review method for the combination of results from independent systematic reviews. Therefore, as an umbrella scoping review, unlike a standard umbrella review, the aim will be to produce a synthesis of the data to date rather than a methodological critique of their quality. The guidance for the Preferred Reporting Items for Systematic Reviews and Meta analyses (PRISMA) protocol for umbrella reviews and Joanna Briggs Umbrella review methodology will be consulted.[29]

Any quantitative papers on social cognition which have been published since the dates of the original systematic reviews will be assessed in the style of the umbrella scoping review. To provide consistency with the quantitative studies in the original systematic reviews (and meta-analysis), a limit for inclusion will be the use of a comparator in their design and have used tests of social cognition with HD cohorts.

Mixed-methods designs on social cognition will not be subject to this latter exclusion. Search terms and keywords for the umbrella scoping review will include those from the individual systematic reviews as well as descriptors relevant to the broader domains of social behaviour and functioning (please refer to box 1).

(2) To explore and summarise the literature related to social behaviour and functioning in HD, guidance which includes the using PRISMA extended protocol for scoping reviews[30] plus the Joanna Briggs Institute method for mixed reviews.[31] Search terms and keywords from the umbrella scoping review will be used.

This section of the review will also include both grey literature and grey data. For grey literature, its appraisal will refer to the most recent best practice guidelines.[32]

For grey data, its use and analysis is considered more challenging[33] and at the current time as there are no specific guidelines, its scope will follow the same cautious

## Box 1 Search terms and keywords

Huntington* AND
Social cogniti* / Social function / Social behaviour / Social interaction / Social judgement / Social psychology / Social understanding / Social skills / Social competence / Social communication / Social decision making
Social withdrawal / avoidance / disengage
Relationships / interpersonal relationships
Social networks / social connections
Work
Sex*
Non-verbal communication
Body language
Facial expression
Recognition
Theory of mind (TOM) / simulation theory
Empath* / empath* communication
Mentalizing / mental state
Emotion*
Inference / Intuition
Function / Skills

## Box 2 Summary of eligibility and inclusion/exclusion criteria

Inclusion
All elements of the review will consider sources of English language evidence which report on adults who are over the age of 18 who are gene positive, presymptomatic or have manifest Huntington's disease (HD). Social cognitive reviews will be defined as: reviews which has used a systematic review approach and included all quantitative and mixed-methods studies which have a comparator group and have the primary outcome using tests of social cognition in people with HD (ie, tests of emotional recognition and/or tests of theory of mind).
The social cognitive review date search will start from 2003 which marks the earliest known social cognitive review by Elamin et al.[20] This will be the start date for all elements of the total review. The final date for all elements of the total review will be up to and including June 2023.
The updated social cognition scoping section will use the latter parameters for inclusion (quantitative, mixed methods, comparator group and social cognitive tests).
The scoping review for social functioning and behaviour will use the definition of social functioning as defined by Kennedy and Adolphs (see earlier comments).
The scoping review for social functioning and behaviour will include any qualitative, mixed or quantitative paper which comments on functioning in people with HD which takes place in a social context and that comments on social interaction or activity which has social consequences. Any disagreement on inclusion will be through consultation between the primary author and supervisor HR.
Search terms and keywords can be found in box 1. These were formed following a review of thesaurus terms and abstracts using two major databases (Medline and Web of Science) using the combination of words—social cognition AND Huntington*
Exclusion
As the protocol forms part of a PhD project, resources and timelines will limit the scope of the literature search and for this reason non-English language evidence will be excluded.

approach applied to grey literature. In practical terms, this means focussing the search of grey literature/grey data to key source material which will be defined in the inclusion/exclusion criteria. Furthermore, in grey data searches, a key consideration is the ownership of the material and therefore its stewardship.[34] While the data are publicly available, there is still the question of its potentially sensitive nature and therefore for ethical reasons and for pragmatic purposes, the data extraction from the source of the grey data will be presented as anonymised and in broad narrative concepts in line with Coulson's use of Cutrona & Suhr's coding system for online bulletin boards. Guidance was also taken from the British Psychological Society's framework on internet mediated research[35] and ethical approval sought and granted through the primary author's University Ethics Review Board (ERN_21-1028A).

(3) To show the organisation of findings using both a mixture of both visual and narrative synthesis in line with current best methodological practice. See analysis and synthesis plan below.

## INFORMATION SOURCES
### Search strategy
The searches will be conducted in five stages (please see online supplemental file 1 for full strategy)
1. The umbrella scoping review of pre-existing systematic reviews starting with Elamin et al.[20]
2. A further (non-umbrella) scoping review of quantitative and mixed methods literature on social cognition, social behaviour and social functioning to the current date. The timeline for the search will run from the first paper annotated in Elamin's review (2003) up to the current date (June 2023). The databases to be used will be Pubmed, Medline, PsycINFO, Web of Science, Scopus, Embase and Cinahl.
3. A scoping review of the qualitative literature concerning social cognition, social behaviour and social functioning. This will match the dates of the pre-existing systematic reviews and run up to the current date. The databases to be used will be the same as for point 2.
4. A grey literature scope using Grey Guide, Google Scholar and Researchgate. The timeline for the search will be the same as at points 1–3.
5. A grey data scope using a UK HD charity affiliated online network. In line with the agreed ethics approval and to reinforce anonymity but not abandon the replicability and therefore rigour of the review, the criteria for this element of the review will not be published. Readers of the review may request the inclusion/exclusion criteria and keywords for the grey data search by writing to the author. Inclusion criteria will use similar parameters as per the scoping review for social functioning (see box 2), that is, comment on relationships.

## Data extraction and data summary
### Qualitative, quantitative and mixed-method studies

In line with the recommended PRISMA guidelines[30] particularly to show the conduct (and therefore rigour) of a review, data relevant to the aims of the review will be extracted from the data sources and 'charted' using an extraction form to summarise the sources' main findings. While this is not a systematic review and therefore methodological quality of the sources is not the main aim, quality cannot be disregarded and in order to optimise the organisation of the synthesis (and thus quality) the Critical Appraisal Skills Programme tool forms matched to the method of the paper (ie, qualitative) will be used as the extraction forms.[36]

### Grey literature

Review of the grey literature will use the Authority, Accuracy, Coverage, Objectivity, Date and Significance checklist designed by Jess Tyndall of Flinders University in 2010 and accepted as the baseline of good review practice.[32]

### Analysis and synthesis

As Popay et al[28] noted, in a review, analysis and synthesis co-exist in that evidence synthesis involves some form of analysis, for example, the finding of common themes across studies or the 'interpretation of results from multiple studies, with the aim of producing new knowledge/findings'.

Popay goes on to state that these different types of synthesis can be located along a continuum from quantitative approaches to qualitative approaches and as recommended in best practice literature the narrative synthesis is the bridge between all of these with other synthesis approaches used in an iterative way depending on the heterogeneity of research designs and findings.

Therefore—after the data extraction, the data from each phase of the search strategy will be entered into their own matrix (using Excel) which will allow for the collation and therefore review of key themes. This framework method which comes from the family of content and thematic analysis methods[37] will lead to the production of the following:

1. An umbrella scoping review of social cognition will be presented through a textual description (alongside a tabular form) in clustered groupings (illustrating types of studies) with relationships between and across studies in a narrative and visual form.
2. A scoping review of social behaviour and functioning which will include the qualitative and grey literature as well as the grey data will also be presented in a tabular form with the main concepts (and relationships) described in a narrative and visual form.
3. A final synthesis will be in a narrative form of key points but also illustrated with a visual concepts map.[38]

## Public and service user involvement

This protocol is part of a wider PhD study on social cognition in HD using ethnography as a method. As the engagement of the HD community was of primary importance, the PPI group of the HDA was approached to comment on the method to see if it would be possible and acceptable but also whether the outcomes would be of value to them.

## Ethics and dissemination

The PhD study which forms the background to this protocol has received ethical approval from the University of Birmingham's STEM ERB with a further addendum specific to this protocol also approved for the internet mediated search of grey data.

The HD community will continue to be involved throughout the life cycle of the study with regular updates at HDA family events and participants will receive personal updates and summaries as requested. The final study will form a standalone piece on the HDA's website.

## DISCUSSION

Social cognition is a set of fundamental skills which are thought to enable successful social interaction. People with HD are known to have both problems with the brain structures associated with social cognition and when tested show this. The tests often take place in controlled settings. HD scientists and neuroscientists want to know more about what happens with social cognition in real life.

This protocol represents the plan for the weaving together of the information we already know from controlled settings alongside of additional observations from the scientific literature on socially related activity in HD as well as evidence from grey literature and grey data. This is intended to form a holistic map of the state of our knowledge of social cognition in HD so that gaps in that knowledge can be targeted; but also, that the knowledge from the review can inform clinical practice.

**Author affiliations**
[1]Institute of Applied Health Research, University of Birmingham, Birmingham, UK
[2]West Midlands Huntingtons Disease Service, Neuropsychiatry, Birmingham and Solihull Mental Health NHS Foundation Trust, Birmingham, UK
[3]Clinical & Experimental Medicine, University of Birmingham College of Medical and Dental Sciences, Birmingham, UK

**Contributors** The final version was read and approved by all authors (AF, AL, SG and HR)

**Funding** The authors have not declared a specific grant for this research from any funding agency in the public, commercial or not-for-profit sectors.

**Competing interests** None declared.

**Patient and public involvement** Patients and/or the public were involved in the design, or conduct, or reporting, or dissemination plans of this research. Refer to the Information sources section for further details.

**Patient consent for publication** Not applicable.

**Provenance and peer review** Not commissioned; externally peer reviewed.

responsibility arising from any reliance placed on the content. Where the content includes any translated material, BMJ does not warrant the accuracy and reliability of the translations (including but not limited to local regulations, clinical guidelines, terminology, drug names and drug dosages), and is not responsible for any error and/or omissions arising from translation and adaptation or otherwise.

**ORCID iDs**
Alexandra Fisher http://orcid.org/0000-0002-6263-6878
Sheila Greenfield http://orcid.org/0000-0002-8796-4114

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
