## [Reviewer comments · BMJ Open]

ARTICLE DETAILS

TITLE (PROVISIONAL)	What does social cognition look like in everyday social functioning in Huntington's disease? - A protocol for a scoping review to explore and synthesise knowledge about social cognition alongside day-to-day social functioning of people with Huntington's disease
AUTHORS	Fisher, Alexandra; Lavis, Anna; Greenfield, Sheila; Rickards, Hugh

VERSION 1 – REVIEW

REVIEWER	Gibson, Jessie S University of Virginia School of Nursing I authored a scoping review of social withdrawal in HD that is referenced in this manuscript.
REVIEW RETURNED	04-Apr-2023

GENERAL COMMENTS	This is an important area where an updated review is warranted. I applaud the authors' plans to also account for impacts on social functioning (i.e., real-life relevance of social cognitive deficits observed in controlled settings). The plans for the review are generally sound, but for reproducibility, more detail is needed regarding the search and selection plans, inclusion criteria, and how concepts of interest will be defined and operationalized (see below). This manuscript could also benefit from careful grammatical revision to avoid run-on sentences and misplaced modifiers. Consistent formatting of headings vs subheadings could also help orient the reader and improve readability. Finally, given the large scope of this project, I wonder about the feasibility of synthesizing the results RE social cognition and social functioning within a single manuscript, if that is indeed the plan. Please see specific comments below: Abstract  • P. 3 line 7, remove "The use of" Introduction/Social cognition in HD  • In general, this manuscript could be strengthened by a bit more background on social cognition in HD, possibly mentioning some the actual findings from the previous reviews/meta-analysis. • P. 3 line 23-25. Recommend clarifying that these problems are thought to manifest prior to motor onset, specifically. Evidence is less clear for timing of onset related to psych/cognitive symptoms. • P. 3 line 40: "...the literature suggests that it is present within crises..." What is "it" referring to? Please clarify. • Please write out acronyms before you first use them (p. 3 line 53). • Recommend adding more detail about the ethnographic study to which you refer on p.3 line 54 and clarify what the "it" is (line 54) that has "inherent and much needed value". Addressing this disparity?
---

	Developing more HD specific measures? This scoping review specifically?  • The authors make a good argument for inclusion of grey literature and the pros and cons of its inclusion in this type of review. Plan  • P. 5 lines 39-40: “ In Grant and Booth’s typography of review methods (29), the authors note that the type of review determines the choice of the review.” Please clarify what you mean by choice of the review. • P. 6 lines 6 and 11. Here, please insert “(Table 1)” to direct the reader to the Search terms table on p. 7. • The methods for search/inclusion/selection of sources of evidence need additional detail. If you include all articles identified with your search terms (p. 7) and meeting those inclusion/exclusion criteria on p. 6, there will likely be a lot of irrelevant articles. You do note previously (p. 5 line 54) that “Any quantitative papers on social cognition” will be included in the social cognition review. Does that mean studies with social cognition as a primary outcome? Secondary outcome? How are you defining social cognition (does that include concepts like emotion recognition, facial affect recognition, etc.)? Who will be making these decisions and resolving disagreements?  o Same concerns for the social functioning review- how are you defining this and how will you determine what is included? o Related to the umbrella review, please specify your plans for making sure you identify all relevant existing reviews of social cognition in HD • P. 7 line 6, I understand the desire to maintain participant anonymity and limitations accepted to access the data, but are you able to provide even a general comment about inclusion criteria? (e.g., “with patient inclusion/exclusion criteria following similar parameters used in previous scoping reviews noted above..”) • The Search terms table (p. 7) should be titled Table 1, so that you can refer to it earlier in the text (see previous comment).
--	---

REVIEWER	Kulisevsky, Jaime Movement Disorders Unit, Neurology Department, Hospital de la Santa Creu i Sant Pau, Universitat Autònoma de Barcelona
REVIEW RETURNED	09-May-2023

GENERAL COMMENTS	This is a well-designed protocol that addresses a topic - social cognition in Huntington's disease - where it is important to increase and disseminate knowledge. The exclusion of non-English languages is a limitation recognized by the authors. The methodology is suitable for this type of meta-analysis. On page 4 'Popes' should be corrected by 'Pope'.
--

VERSION 1 – AUTHOR RESPONSE

Reviewer 1 comments

In the abstract – the line ‘the use of’ has been removed

In the introduction - there is increased detail on social cognition, what it means and how it is measured. The outcome of the current social cognition reviews in HD have been mentioned. That social cognitive problems manifest prior to motor symptoms has been clarified and referenced. ‘It’ has

been clarified as social cognitive functioning problems. Acronyms have all been written out in full before using them. There is more detail about the ethnographic study. 'It' has been clarified as the study of social cognitive problems in real life.

In the plan – the comment by Grant and Booth has been removed as it is not needed. Readers have been directed to the search terms table. Inclusion and exclusion criteria have been made clearer and summarised in the form of a new table. Arrangements for disagreements have been made. The operationalisation of social functioning has now been included in the introduction and is signposted to in this section with examples given. Inclusion/exclusion criteria as to how all social cognitive reviews will be included is embedded within this section also. A general comment has been given about the grey data search has been made.

Reviewer 2 comments

Popes has been changed to Pope

Many thanks for your further consideration of my revised submission.

VERSION 2 – REVIEW

REVIEWER	Gibson, Jessie S University of Virginia School of Nursing
REVIEW RETURNED	26-Jun-2023

GENERAL COMMENTS	Thank you for the opportunity to review this revision. It is much improved by the updated introduction, Table 1, and search protocol supplement.
--